# In Silico Identification and Functional Characterization of Genetic Variations across DLBCL Cell Lines

**DOI:** 10.3390/cells12040596

**Published:** 2023-02-12

**Authors:** Prashanthi Dharanipragada, Nita Parekh

**Affiliations:** Center for Computational Natural Sciences and Bioinformatics, International Institute of Information Technology, Hyderabad 500032, India

**Keywords:** diffuse large B-cell lymphoma (DLBCL), copy number variations, sequence variations, structural variations, cell lines, precision medicine

## Abstract

Diffuse large B-cell lymphoma (DLBCL) is the most common form of non-Hodgkin lymphoma and frequently develops through the accumulation of several genetic variations. With the advancement in high-throughput techniques, in addition to mutations and copy number variations, structural variations have gained importance for their role in genome instability leading to tumorigenesis. In this study, in order to understand the genetics of DLBCL pathogenesis, we carried out a whole-genome mutation profile analysis of eleven human cell lines from germinal-center B-cell-like (GCB-7) and activated B-cell-like (ABC-4) subtypes of DLBCL. Analysis of genetic variations including small sequence variants and large structural variations across the cell lines revealed distinct variation profiles indicating the heterogeneous nature of DLBCL and the need for novel patient stratification methods to design potential intervention strategies. Validation and prognostic significance of the variants was assessed using annotations provided for DLBCL samples in cBioPortal for Cancer Genomics. Combining genetic variations revealed new subgroups between the subtypes and associated enriched pathways, viz., PI3K-AKT signaling, cell cycle, TGF-beta signaling, and WNT signaling. Mutation landscape analysis also revealed drug–variant associations and possible effectiveness of known and novel DLBCL treatments. From the whole-genome-based mutation analysis, our findings suggest putative molecular genetics of DLBCL lymphomagenesis and potential genomics-driven precision treatments.

## 1. Introduction

Diffuse large B-cell lymphoma (DLBCL) is the most aggressive form of non-Hodgkin’s lymphoma, demonstrating high molecular and clinical heterogeneity. Several classification systems have been proposed on the basis of the shared morphology, immunophenotype, clinical outcomes, etc., in order to stratify patients, decipher mechanisms of pathogenesis, and design suitable therapy [1,2,3,4,5]. The Ann Arbor staging classification system, on the basis of the number and localization of malignant lymphatic sites, categorizes patients into four stages (I-IV) [3]. Similarly, for prognosis purposes, the International Prognostic Index (IPI) is computed on the basis of risk factors, age, serum lactate dehydrogenase (LDH) levels, stage III-IV, performance status, and the number of extra-nodal sites [4]. Gene expression profiling of DLBCL patients has led to the identification of two distinct subtypes on the basis of the cell-of-origin (COO) classification, the germinal center B-cell (GCB) and activated B-cell (ABC) subtypes [5]. A third ‘unclassified’ subtype is observed in 10–20% of DLBCL tumors that do not fall into either GCB or ABC subtypes (a few cases possibly belong to primary mediastinal B cell lymphoma (PMBL)). Poor clinical outcome in ABC compared to GCB samples is indicative of molecular heterogeneity because of their genomic profile [6].

Genetic variations, majorly categorized as small sequence variations (single-nucleotide variations and short insertion-deletions) and structural variations (copy number variations (CNVs), inversions, translocations, and mobile element insertions (MEIs)), are often associated with the initiation, progression, and varied outcomes in DLBCL [7,8]. Small sequence variations (SSVs) are known to affect different molecular pathways through gain-of-function of proto-oncogenes or loss-of-function of tumor suppressors in the course of B-cell development and differentiation. Within germinal B center (GC) B cells, immunoglobulin variable (IGV) regions undergo mutations at a higher rate (×106 higher compared to normal somatic mutations) to generate diverse antibodies in a natural phenomenon termed somatic hypermutation. However, aberrant somatic hypermutation (aSHM) may target non-IGV genes such as BCL6, MYC, FAS, and PAX5, leading to genome instability and, subsequently, lymphomagenesis. This process is highly specific to DLBCL, affecting >50% of DLBCL cases and being rarely observed in other B-cell malignancies [7]. Earlier studies have also reported distinct sets of SSVs in a subtype-specific manner.

Structural variations (SVs) are broadly classified into balanced (translocations and inversions) and unbalanced variations (insertions, duplications, and deletions). Despite balanced structural variations being generally harmless, the heterozygous condition of these variations is shown to result in genetic abnormalities and tumorigenesis. Recombination of VD(J) regions is another key event in antigen receptor loci of B cells and malfunction of the process that plays an important role in the generation of chromosomal translocations. Translocations of proto-oncogenes BCL6 and MYC (‘double-hit’) are observed in ≈10% of DLBCL cases and are categorized as ‘high-grade B-cell lymphoma’ by the WHO. These translocations are associated with deregulated cell cycle and faulty DNA repair mechanisms, leading to uncontrolled growth of lymphoma cells [9]. Another important type of structural variation is mobile element insertions (MEIs), which are DNA sequences that can move around the genome, changing their number of copies or simply changing their location, and in the process may affect the activity of nearby genes. These include SINES (mostly Alu repeats), LINES (mostly LINE1-L1), and SVA (SINE-R/VNTR/Alu) elements observed in various types of cancers. MEIs are well known to generate new MEIs and other structural variations in the vicinity. The role of unbalanced variations such as CNVs is well studied in DLBCL [8,10]. Amplification of oncogenes FCGR2B and ARID2 and deletion of tumor suppressor genes CIITA and TP53 have been identified to significantly play a role in lymphomagenesis through disruptions in different immuno-oncogenic pathways. Copy gain of BCL6, MDM2, and REL and copy loss of PTEN, FAS, and ING1 are particularly observed in GCB-DLBCL patients, while copy gain of SPIB and FOXP1 and loss of PRDM1 and CDKN2A are frequently observed in ABC-DLBCL patients [11].

Although the majority of DLBCL patients (≈60%) are cured with an R-CHOP immunochemotherapy regimen, a significant number of remaining patients do not benefit from it. Conversely, for a fraction of patients, it is refractory, whereas the rest succumb to progressive/relapsed disease. The difference in the outcome is indicative of the difference in the genetic variation profile of the patients. Recently, novel classifications of DLBCL subtypes have been proposed on the basis of genetic variations or including transcriptomic expression profiles. These studies have employed classification approaches such as random forest [12] or non-negative matrix factorization (NMF) consensus clustering [8]. For instance, genomic and transcriptomic analysis of 574 DLBCL biopsy samples revealed four distinct subtypes, viz., MCD (co-occurrence of MYD88L265P and CD79B mutations), BN2 (BCL6 fusions and NOTCH2 mutations), N1 (NOTCH1 mutations), and EZB (EZH2 mutations and BCL2 translocations) [12]. Another study [8] on a cohort of 304 primary DLBCL cases identified five distinct molecular subsets (C1-C5) on the basis of genetic variations, as well as an additional GCB/ABC independent group (C0) without detectable genetic alterations. These subtypes exhibit distinct molecular profiles, activation/inhibition of different pathogenic pathways, and varied response to immuno-chemotherapy, affecting the overall outcome of patients. Clearly, due to the high genetic complexity observed, traditional COO classification categorizing patients into GCB or ABC subtypes is not enough for patient stratification and recognizing potential targets for new intervention strategies.

Copy number aberrations are common in DLBCL (cancer, in general) and have a very high impact due to their direct effect on oncogenes and tumor suppressor genes, as discussed in our earlier work [13]. However, SSVs and other SVs are also known to play an important role, as seen from various studies [14,15,16], and continue to emerge in the understanding of complex diseases. Characterizing genetic variants would allow us to understand the biology of DLBCL pathogenesis through identifying genes involved in rearrangements, fusion events, and the mobilization of gene/regulatory elements, in addition to truncated or amplified genes (as seen in CNVs). Each of these events have been identified in the pathogenesis of various types of cancers including DLBCL [14]. This clearly indicates the need for integrating various genetic variants identified from whole-genome sequence data to obtain a comprehensive profile of all possible genetic alterations. Additionally, this acts as a first step towards identifying and targeting patient-specific immuno-oncogenic pathways. In this work, we discuss the detailed analysis of genetic variations in eleven DLBCL cell lines (seven GCB and four ABC) in order to identify affected pathways and drug responses. The key objectives of the analysis are to identify functionally significant variants and integrate SSVs with CNVs and other SVs identified in each cell line to understand their collective role in DLBCL lymphomagenesis.

## 2. Materials and Methods

The data comprised paired-end sequencing of lymphoma cell lines (7 GCB and 4 ABC subtypes) with a read length of 100 bp using an Illumina HiSeq 2000 platform (Appendix A). The whole-genome sequences of the DLBCL cell lines considered for analysis in this study were generated as part of the National Cancer Institute (NCI) and Cancer Genome Characterization Initiative (CGCI): Non-Hodgkin Lymphoma project, dbGAP accession: phs000532.v16.p5; SRA project ID: SRP020237 [17]. The cell lines were obtained from Leibniz-Institut DSMZ (DB, DOHH-2, NU-DHL-1, NU-DUL-1, SU-DHL-6, and WSU-DLCL2); the U.S. National Institutes of Health (OCI-LY1, OCI-LY7, and OCI-LY19); and the University of Leicester, UK (MD903, SU-DHL-9). The paired-end reads with base quality (<30) were trimmed using NGS QC-Toolkit [18]. Trimmed reads were then aligned to human reference assembly hg19 using Bowtie2 [19]. Aligned files were sorted and split chromosome-wise using SAMtools [20]. The detection of small sequence variants (SSVs), i.e., SNVs and short INDELs (≤50 bp), was carried out using our in-house tool, Sequence Variants Identification and Annotation (SeqVItA) [21]. It is an open-source integrated platform that provides an end-to-end solution from pre-processing of data to variant calling, as well as variant annotation and prioritization in whole-genome (WGS), whole exome (WES), and targeted (TS) next-generation sequencing (NGS) data. It is based on a heuristic approach for variant calling and is parallelized using OpenMP for computational efficiency. Numerous resources have been incorporated to infer functional impact, clinical relevance, and drug-variant associations for the variants called. Reads with mapping quality <20 were discarded, and SSVs were identified using default parameters. Translocations (TRA) and inversions (INV) from the cell lines were predicted using TIDDIT [22]. TIDDIT allows for the detection of an entire spectra of SVs, namely, translocations, inversions, deletions, interspersed duplications, insertions, and tandem duplications. It uses discordant reads, split reads, and read depth information for detecting the genomic location of SVs. Mobile insertion elements (MEIs) were identified using the Mobile Element Locator Tool (MELT) [23]. It uses reference MEs, discordant reads, and split-read signals for the precise prediction. Genetic variants identified included both germline and somatic mutations. The CNVs were detected using our in-house tool, iCopyDAV [24]. It is an integrated open-source platform for the detection, annotation, and visualization of CNVs in WGS data. It is based on the depth-of-coverage approach and does not require a matched-control for CNV prediction. Parallelization of the segmentation step and functional annotation and prioritization of the predicted CNVs are its important features. From the predicted list of SSVs, SVs, and MEIs, those reported in the 1000 Genome Project (aligned to hg19 human reference assembly) were filtered out to screen germline mutations as these are not likely to contribute towards tumorigenesis. The remaining variants (both germline and somatic) were considered for further analysis.

Functional annotation and prioritization of SSVs identified in DLBCL cell lines was carried out by the annotation module in SeqVItA. It provided annotations in three categories, viz., functional impact (phylogenetic conservation scores from SIFT, PolyPhen2, LRT, MutationTaster, and PhyloP), clinical relevance (ClinVar, COSMIC, OMIM, and DECIPHER), and variant/gene–drug associations (PharmGKB). Oncogenes (OGs) and tumor suppressor genes (TSGs) containing ‘high’ and ‘medium’ priority sequence variants and overlapping structural variants were identified using the Molecular Signatures Database (MSigDB) [25]. Validation of SSVs and prognostic significance was carried out using 1295 DLBCL samples in cBioPortal for Cancer Genomics [26]. Co-occurrences (or mutual exclusivity) of genetic alterations in cBioportal were computed by calculating the odds ratio that indicated the likelihood that the events in the two genes were mutually exclusive or co-occurrent across the selected cases. Fisher’s exact test was computed to identify whether the identified relationship was significant [26]. Genes overlapping INVs and TRAs were extracted using AnnotSV [27], and key genes (OGs and TSGs) were obtained from MSigDB. AnnotSV is a standalone resource for annotation and ranking of SVs. It provides genomic-based annotations such as overlap with Refseq genes, promoters, DGV gold standard, deciphering developmental disorders (DDD), the 1000 genome project, gene intolerance, haploinsufficiency, OMIM, dbVar, GC content, repeats, and topologically associating domains in different subtypes of SVs. Annotation of CNVs was carried out using the annotation module of iCopyDAV and included functional (genes, enhancers, lincRNA, miRNA target sites), clinical (clinVar, OMIM, DECIPHER, ExAC), and structural (segmental duplications from WGAC, tandem repeats from TRFinder, and interspersed re-peats from RepeatMasker) annotations. Functional enrichment analysis of the key genes was carried out using STRING [28] and g:Profiler [29]. Finally, the combined impact of the genetic variants, namely, SSVs, CNVs, INVs, MEIs, and TRAs, was assessed to understand their role in lymphomagenesis and to identify hotspots of genome instability in the cancer genome.

## 3. Results

### 3.1. Distribution of Sequence and Structural Variants

The number of SSVs, CNVs, INVs, TRAs, and MEIs identified across the 11 DLBCL cell lines are depicted in Figure 1 as a component bar graph, while the number of SSVs was given by the absolute value of ‘×’. As expected, the number of SSVs observed was very large (≈2400 per cell line). This was followed by the number of translocations (≈1000 per cell line), MEIs (≈890), CNVs (≈350), and inversions (≈60). The number of CNVs, SSVs, ALU, and SVA were observed to be proportional to the size of the chromosome, with the highest number of these variations in chromosome 1 (Appendix A). However, no such trend was observed in the distribution of inversions (highest in chr2), translocations (chr4), and LINE elements (chr9). Among MEIs, a large proportion comprised ALU elements (≈80%), with the remaining amount being LINEs (10–15%) and SVAs (5–10%) lineages, as shown in Figure 2.

### 3.2. Key Altered Oncogenes and Tumor Suppressor Genes in DLBCL Cell Lines

To assess the functional impact of SSVs and SVs in DLBCL lymphomagenesis, oncogenes and tumor suppressors encompassing these genetic variants were extracted using MSigDB [25] and are briefly discussed below.

#### 3.2.1. Small Sequence Variations

Oncogenes and tumor suppressor genes containing ‘high’ and ‘medium’ priority mutations were extracted (Appendix A), and those present in one or more cell lines are shown in Figure 3. Recurrent mutations were observed in genes TP53, FCGR2B, BCL6, BCL2, MYC, and EGFR, which were implicated in DLBCL in earlier studies [30]. Mutations in genes BCL2 (6/11 cell lines), MYC (4/11), BCL6 (3/11), POU1AF1 (2/11), and PIM1 (1/11) are known to be potential targets of aberrant somatic hypermutation (aSHM) [7]. Of these, mutations in PIM1 and POU1AF1 were observed to be specific to ABC cell lines, while mutations in BCL2, MYC, and BCL6 were present in both the subtypes. Some genic SSVs, although observed in only a few cell lines, may play an important role in DLBCL lymphomagenesis. For example, mutations in CREBBP were observed in only two GCB cell lines, NU-DHL-1 and WSU-DLCL2. Gene CREBBP encodes for transcriptional activator CREB-binding protein and is involved in histone acetylation. Studies on an in vivo mice model with mutations in CREBBP were shown to exhibit decreased histone H3 acetylation, reduced MHC II expression, and increased cell proliferation compared to the wild type [31]. Similarly, mutations in genes CARD11 and PIM1 were observed in only one ABC cell line, MD903. Interestingly, from the data in cBioPortal, mutations in CARD11 and PIM1 tended to co-occur in 29/1295 DLBCL patients with high significance (Fisher’s exact test, *p*-value: 0.03).

#### 3.2.2. Copy Number Variations

We carried out an extensive functional analysis of CNVs in the 11 DLBCL cell lines in our previous work [13]. Briefly, a highly heterogenous distribution of CNVs was observed across the cell lines. Key genes were extracted using MSigDB, and on an average, ≈10 oncogenes and ≈2 tumor suppressor genes per cell line were observed to lie on copy gain and copy loss regions, respectively (Appendix A). Except for a few CNV genes such as NOTCH2 and HLA-I/II, the majority of CNV genes were present in only a few cell lines, in a subtype-independent manner. Analysis of these CNV-enriched key genes revealed possible mechanisms of DLBCL lymphomagenesis in different cell lines. For example, deletion of the tumor suppressor PTEN was identified to trigger the PI3K-AKT-mTOR signaling pathway responsible for tumor growth and survival in GCB-DLBCL cell lines OCI-LY1 and WSU-DLCL2. Conversely, in GCB cell lines NU-DHL-1 and SU-DHL-6, amplification of cyclin proteins CCND2 and CCND3 oncogenes were found to be associated with uncontrolled growth of cancer cells. Five novel CNV genes, ERICH1, DLEU1, BMPR1A, DEK, and SUFU, were proposed as prognostic markers due to their poor survival in patients compared to those without CNVs in TCGA. 

#### 3.2.3. Translocations

Genomic rearrangements through translocations were the most common type of genomic alterations detected in DLBCL. We observed both subtype-specific and commonly shared translocations between the GCB and ACB cell lines. Translocations between immunoglobulin heavy-chain (IgH) and BCL2 were observed in four GCB cell lines (OCI-LY1, DB, DOHH-2, and SU-DHL-6) in agreement with the cytogenetic results reported in the DSMZCellDive web portal [32], and between IgH and BCL6 in one ABC cell line (MD903). On the other hand, cell lines NU-DUL-1 (ABC), OCI-LY7 (GCB), and DOHH-2 (GCB) exhibited well-known MYC rearrangements t(8;14). A dual translocation (t(14;18) + 8q24) involving BCL2-IgH and MYC, generally observed in rare DLBCL cases and characterized by aggressive clinical presentations, was observed in the DOHH-2 (GCB) cell line (Table 1 and Appendix A). Additionally, a few novel fusions of oncogenes resulting from translocation events were identified (summarized in Table 1). To the best of our knowledge, these novel fusions are not yet reported in DLBCL. Some of the key inter- and intra-chromosomal translocation events observed in the 11 cell lines along with associated genes and potential significance are listed in Table 1.

#### 3.2.4. Inversions

Limited studies have been carried out to identify and characterize inversions in DLBCL genomes probably because inversions are balanced variations and generally do not cause abnormality in the function of gene(s) spanning them. Nevertheless, a heterozygous inversion would result in the formation of abnormal chromatids and subsequently affect the gene expression. In our analysis, we observed several OGs and TSGs (≈41 genes per cell line, median: 37, range: 9–88) altered due to heterozygous inversions (Appendix A). A large variation in the number of transversions was observed, ranging from 87 key genes disrupted in the GCB cell line NU-DHL-1 to as few as 9 genes in the GCB cell line SU-DHL-6. A total of 99 unique key genes were identified across the 11 DLBCL cell lines, of which at least 44 genes were inverted in any three DLBCL cell lines and 64 genes in any two cell lines (Appendix A). Oncogenes ABL2 and PRRX1 were observed to be altered due to inversions across all the eleven cell lines.

The gene ABL2 is involved in cell growth and survival through cytoskeleton remodeling, while the gene PRRX1 enhances the DNA-binding activity of serum response factor, a protein required for the induction of genes by growth and differentiation factors. A large inversion of size 20 Mbp was observed at 11q12-q14 in the five DLBCL cell lines OCI-LY1, NU-DHL-1, SU-DHL-6, DOHH-2, and SU-DHL-9 and encompassed three oncogenes (CCND1, MALAT1, NUMA1) and two TSGs (MEN1, SDHAF2). Among these, abnormal expression of CCND1 due to duplications/genomic rearrangements is previously reported in DLBCL [33], while higher expression of lncRNA MALAT1 was associated with tumorigenesis and immune escape in DLBCL [34]. Another large inversion of the size ≈99 Mbp at 12q arm was observed in the four GCB cell lines OCI-LY1, DB, DOHH-2, and WSU-DLCL2. This chromosomal region was enriched with eleven OGs and one TSG. Interestingly, 6/11 oncogenes (HMGA2, DDIT3, CDK4, PTPN11, MDM2, and BTG1) were directly involved in the negative regulation of the cell cycle (*p*-value: 9.77 × 10^−9^).

#### 3.2.5. Mobile Element Insertions

Mobile element insertions, another common form of structural variations, were also observed in DLBCL cell lines. As shown in Figure 2, a majority of these were ALU elements. Although several genes are proportionately altered, the number of key OGs and TSGs modified by ALU are very few, typically four per cell line. Some key genes overlapping with ALU elements (and a few SVA and LINEs) and various immuno-oncogenic pathways dysregulated due to MEIs are listed in Table 2.

### 3.3. Genetic Variations Disrupt Key Immuno-Oncogenic Pathways

The number of OGs and TSGs were observed to overlap various genetic variations and may lead to the disruption of key cellular processes. To understand their functional impact in contributing to DLBCL development and progression, we investigated the role of these key genes (containing variations) using STRING and g:Profiler. Common immuno-oncogenic pathways, namely, B-cell receptor (BCR), cell cycle/apoptosis, JAK-STAT, MAPK, NF-κB, PI3K-AKT, and receptor tyrosine kinase (RTK), involved in cell growth, proliferation, and survival, were observed to be affected by these genetic variants. A few of these genes were also associated with pathways that are druggable, e.g., NF-κB, PI3K-AKT, and RTK.

In different individuals, various routes to tumorigenesis were observed—different genetic variants leading to the disruption of the same pathways, or disruption of different genes/pathways and cross-talk between pathways is common. For example, in OCI-LY1, we observed 16 key genes (9 OGs and 7 TSGs) solely affected by CNVs, and on considering other genetic variations, ≈70 additional key genes that possibly participate in DLBCL pathogenesis were observed. This clearly suggests a need for integrating SSVs with CNVs and other SVs to obtain a complete variant profile of a patient for a better understanding of the DLBCL pathogenesis and to be able to prescribe individualized diagnosis and treatment. Different types of genetic variations in essential genes affecting various immuno-oncogenic pathways involved in lymphomagenesis across the 11 DLBCL cell lines are summarized in Figure 4. For a few genes, more than one single type of large structural variation was observed (indicated in gold yellow in Figure 4). For example, such complex rearrangements in the NOTCH2 gene were observed in 10/11 cell lines, leading to the disruption of the NOTCH signaling pathway. These rearrangements include copy gain, inversions, and mobile element insertions. Similarly, the genes MYC, CDKN2A, and RAF1 also exhibited complex rearrangements comprising CNVs and translocation events in different cell lines. These aberrations consequently affect the tumor cell growth and proliferation. We also observed several immuno-oncogenic pathways commonly affected by SSVs, CNVs, and SVs in DLBCL cell lines. These include the activation of BCR signaling; the deregulation of cell cycle/apoptosis; and the activation of JAK-STAT, MAPK, NF-κB, NOTCH, PI3K-AKT, and RTK signaling pathways. Certain pathways such as epigenetic modifications and WNT/β-catenin pathways are majorly altered through SSVs in most DLBCL cell lines that cannot be revealed using CNV analysis alone. It would be interesting to assess the gain/loss of function of the key genes due to different genetic variations.

### 3.4. Comprehensive Role of Genetic Variations in Lymphomagenesis

Below, we discuss our analysis by integrating the information of SSVs, CNVs, and SVs encompassing keys genes and show that it can provide insights into molecular differences between the two cell lines, DB (GCB subtype) and MD903 (ABC subtype). On comparing the genetic profile of these two cell lines, a striking difference in SSVs was observed, with mutations in EZH2 affecting the chromatin modifications in DB, while mutations in CARD11 led to constitutive expression of NF-κB signaling pathway in the MD903 cell line. Although most other pathways deregulated in these two cell lines are common (BCR, cell cycle, epigenetic modifications, PI3K-AKT), the genes/genetic variations involved in altering these pathways were different. For example, amplification of oncomir, mir-17-92 in DB and missense mutation in PIK3R1 in MD903 led to the disruption of the PI3K-AKT pathway in both the cell lines. Similarly, mutation/translocations in BCL2, mutations in TP53 and RUNX1 (DB), and copy loss of CDKN2A (MD903) were seen to affect cycle cell progression in the two cell lines. Even within the same subtype, differences in the genetic profile were observed. For instance, although the GCB cell lines DB and OCI-LY7 were identified with the amplification of mir-17-92 that directly affects the PI3K-AKT-mTOR pathway, mutations in different genes EZH2, RUNX1, CCND3, and BCL6 in the DB cell line, and HSP90AA1 and BCL2 in the OCI-LY7 cell line, were observed to affect distinct immuno-oncogenic pathways, as shown in Figure 4. These differences in the genetic variations clearly indicate the heterogeneity in between and within DLBCL subtypes, reflecting varied disease progression and treatment outcomes among the patients.

A more complex picture of the pathways involved in tumorigenesis was observed by considering all the genes affected by SSVs, CNVs, and other SVs. For example, in the OCI-LY1 (GCB) cell line, shown in Figure 5, chromosomes 8, 11, and 12 were identified with several key genes affected by INVs (≈43 genes), and a few by CNVs (≈6 genes) and SSVs (≈3 genes). Some of the key genes affected by INVs and CNVs were PTEN, FOXO1, and RB1 in chromosomes 10 and 13. As a result of CNVs, the PI3K-AKT-mTOR pathway (PTEN), cell cycle (FANCE and FAS), chromatin modifications (KDM5A, PRDM16), JAK-STAT (JAK2), and NF-κB (ERC1) were disrupted in OCI-LY1. Genomic deletion at the PTEN locus in the OCI-LY1 cell line resulting in no detectable PTEN protein expression was shown by Pfeifer et al. [35]. The authors demonstrated that PTEN expression was inversely correlated with AKT phosphorylation (p-AKT) status in GCB DLBCL cell lines. Further, characterizing SSVs in OCI-LY1 revealed alterations, namely, in cell cycle/apoptosis (RUNX1 and TP53), chromatin modifications (EZH2), JAK-STAT (STAT6), B-cell receptor (BCL2), MAPK (NTRK1), RTK (EGFR), and the WNT signaling pathway (APC and SMAD4). Similarly, alterations in cell cycle (BCL2, FAS, FOXO1, MYC), MAPK (RAF1), and PI3K-AKT (FOXO3, PTEN) signaling pathways were revealed through SVs. High basal protein expression of BCL2 was shown in the OCI-LY1 cell line using Western blot analysis by Sun et al. [36], which was also reported by cytogenetics in DSMZCellDive, providing support to our prediction. Certain pathways such as chromatin modifications, cell cycle, and JAK-STAT were affected by both SSVs and CNVs through different genes, providing a strong indication of a disruption of these pathways in the OCI-LY1 cell line. Thus, identifying genes affected by various genetic variants can assist in identifying and better understanding of the molecular pathways affected in patient samples.

Detailed analysis of the genetic profiles of cell lines also revealed possible subgroups within the GCB and ABC subtypes that were based on shared genetic variations. For example, GCB cell lines OCI-LY1 and WSU-DLCL2 shared mutations in BCL2, EZH2, deletion/inversion of PTEN, and FOXO3/PDGFD translocations, and these cell lines may possibly follow a similar course (e.g., PI3K-AKT signaling) to DLBCL tumorigenesis because of their shared genetic profile. Moreover, this subgroup shared similar genetic features as the C3 subset identified by Chapuy et al. [8]. Interestingly, a combination of genetic variations in BCL2, PTEN, and FOXO3 identified in this subgroup showed a lower disease-specific survival in patients (*p*-value: 0.05) compared to those without alterations in these genes reported in 48 patients reported in the TCGA. Furthermore, GCB cell lines NU-DHL-1 and DOHH-2 were identified with driver genetic variation through the loss of CDKN2A, which shared characteristics of the C2 subset reported in the same study. Other pairs of cell lines with shared genetic variants are listed in Table 3 along with the pathways affected.

### 3.5. Diverse Paths of DLBCL Pathogenesis 

On the basis of our analysis of complete genetic variant profiles of 11 cell lines, we observed different routes to the disruption of key immunogenic pathways, as shown in Figure 6. Four cell lines (OCI-LY1, WSU-DLCL2, SU-DHL-6, and NU-DUL-1) demonstrated the activation of the PI3K-AKT signaling pathway through pathogenic genetic variations. The key variations include loss of PTEN, copy gain of HSP90AA1, and mutations in FOXO3 or BCL2, all of which are well established to contribute to tumor cell growth and survival. Briefly, the deletion of tumor suppressor gene PTEN lost control of PI3K, leading to the accumulation of PI3K and activation of a downstream signaling cascade. Oncogene and chaperon HSP90AA1 interacted with AKT and triggered the AKT-mTOR signaling pathway. The FoxO set of transcription factors, downstream of AKT, were regulated by PI3K-AKT and inhibited cell growth through inducing the expression of multiple pro-apoptotic elements. Disruption in their genetic makeup either by mutations in FOXO3 or structural variations in FOXO1 caused inactivation and resulted in the accumulation of FoxOs, enabling cell cycle and progression. 

Alternate mechanisms were observed in four DLBCL cell lines (OCI-LY7, MD903, OCI-LY19, and SU-DHL-9) mainly through mutations in EP300, SMAD4, and APC, resulting in dysregulation of TGF-β signaling and WNT signaling pathways. Upon interaction of β-catenin with various transcription factors of the TCF/LEF family, several trans-activators such as the CREB binding proteins EP300 and BCL9 were recruited for downstream gene expression involved in carcinogenesis. In vitro studies revealed that histone acetyl transferases (HATs) such as EP300 are capable of binding to β-catenin, leading to the activation of the WNT signaling pathway. The gene EP300 also exhibits a dual role of inhibiting and activating BCL6 through acetylation, allowing B cells to rapidly reprogram to adapt to the signal. Mutations in gene BCL6 were observed in DLBCL cell lines. Additionally, EP300 influenced the activity of the tumor suppressor gene TP53, suggesting its role in tumorigenesis through alternate mechanisms [37]. The gene SMAD4, a central mediator of the TGF-β signaling pathway, is involved in several biological processes including cell growth, differentiation, and apoptosis and acts as crosstalk mediator between TGF-β and WNT signaling pathways, which mainly occur in the nucleus. Although loss of function of SMAD4 does not directly initiate tumorigenesis, it can promote tumor progression initiated by other genes, such as APC. Inactivation of the APC gene promotes tumor progression by increasing WNT/β-catenin signaling. Thus, mutations in various genes involved in the WNT signaling pathway indicate a possible alteration of this pathway in DLBCL cell lines. 

In two cell lines (NU-DHL-1 and DOHH-2), genetic alterations were observed to directly influence the cell cycle pathway. Among various functions, MYC predominantly promotes the transition of cells from the G0 to the S1 phase, either through directly regulating the CCNDs or inhibiting cell cycle inhibitors. Genetic rearrangements in MYC result in its overexpression, eventually leading to the dysregulation of the cell cycle. Additionally, we observed a copy loss of TSG CDKN2A and driver mutations in TP53 and CCND3, further enabling the tumor cells to grow.

### 3.6. Pharmacogenomic Implications of Genetic Variants 

We next examined pharmacogenomic implications of the genetic variations observed in DLBCL cell lines. From Figure 4 and Figure 6, it is shown that that only a few key pathways were commonly affected through different types of variations in key OGs and TSGs. For example, constitutive activation of PI3K-AKT signaling is an important mechanism through which tumor B cells grow, proliferate, and survive in DLBCL. It is observed from Figure 4 that the alteration of PI3K-AKT pathway was observed in 10/11 cell lines through various mechanisms: mutations in FOXO3 (DOHH-2, WSU-DLCL2, SU-DHL-6, NU-DUL-6) and PIK3R1 (SU-DHL-6, MD903), CNVs in HSP90AA1 (OCI-LY1, WSU-DLCL2), HSP90AB1 (SU-DHL-6, NU-DUL-1, SU-DUL-9), miR-19-72 (DB, OCI-LY7, SU-DHL-6), PTEN (OCI-LY1, WSU-DLCL2), and RPTOR (MD903, SU-DHL-9), or other SVs in FOXO3 (OCI-LY1, DOHH-2, WSU-DLCL2, OCI-LY19), PIK3R1 (NU-DHL-1, NU-DUL-1), and PTEN (OCI-LY1, OCI-LY7, WSU-DLCL2). Figure 6 shows the probable mechanisms of the PI3K-AKT pathway activated in four DLBCL cell lines according to shared genetic variations. Inhibitors directly targeting PI3K (e.g., PIK3R1) through the use of drugs LY294002, idelalisib, and AZD8835, and AKT through drug AZD5363 and mTOR (e.g., RPTOR) inhibitors such as rapamycin, have proven to be efficient therapeutics in DLBCL patients [9]. Thus, depending on the affected component of the PI3K-AKT-mTOR pathway, an appropriate treatment plan can be designed. From several studies, chronic active BCR and NF-κB signaling pathways are well known to contribute to DLBCL pathogenesis, predominantly in ABC subtype. From Figure 4, activation of BCR and NF-κB signaling pathways were observed through mutations in CD79B (MD903); CARD11 (MD903); CNVs in IRF4 (SU-DHL-6, NU-DUL-1), BCL6 (DB), and ERC1 (OCI-LY1, NU-DHL-1, MD903); and other SVs in BCL6 (MD903) and REL (DB, MD903, OCI-LY19) genes across 8/11 DLBCL cell lines. Drugs such as ibrutinib, lenalidomide, and bortezomib are inhibitors of various components of BCR and NF-κB signaling pathways, and the use of these has been shown to reduce BCR/NF-κB pro-survival signaling in patients [9]. By considering SSVs, CNVs, and SVs, we observe a larger set of cell lines exhibiting disruption of similar pathways, giving us a clearer picture of the biological processes affected because of various types of variations. From these results, it is evident that for targeted therapy, it is important to analyze the immuno-oncogenic pathways affected in patient samples through any type of genetic variation. 

### 3.7. Drug–Variant Associations

Apart from identifying targetable pathways, we observed that certain mutations in genes can also affect the drug response in patients. For example, a missense mutation (A>C) at chr1:161514542 (rs396991) in the FCGR3A gene was identified in nine cell lines (six GCB and three ABC subtypes, except DB and NU-DUL-1 cell lines). Annotations in PharmGKB indicated the association of this mutation with an increase in the efficacy of the rituximab drug. The gene FCGR3A encodes for a receptor of the Fc portion of immunoglobulin G (IgG) and is involved in the elimination of antigen–antibody complexes. It also mediates antibody-dependent cellular cytotoxicity (ADCC), a complex process in which effector cells from the host immune system actively target and kill the cells whose membrane antigens are bound with specific antibodies. Aberrations in this gene are associated with increased susceptibility of recurrent viral infections, autoimmune disease systemic lupus erythematosus, etc. Interestingly, the action of drug rituximab, a monoclonal antibody (anti-CD20), is based on the ADCC mechanism, and the presence of the missense mutation (rs396991) is associated with increased affinity of FCGR3A for IgG. This results in increased efficacy of rituximab. The drug rituximab (R) is combined with the chemotherapy drug CHOP and is widely used in the treatment of DLBCL. In a cohort study on an East Asian population (n = 113), this mutation was shown to be associated with an increased complete response rate in patients treated with R-CHOP compared to those without this mutation [38]. However, a recent meta-analysis study by Ghesquières et al. showed no such correlation between the FCGR3A polymorphism and the overall survival of DLBCL patients using two independent cohort studies (n = 1134, America and Europe study locations) [39]. These contradictory results indicate the possible role of the variant in an ethnically specific manner.

The gene FLT4 encodes for receptor tyrosine kinase (RTK) for vascular endothelium growth factors, involved in lymph-angiogenesis, and promotes endothelium cell growth, proliferation, and survival. A missense mutation at chr5:180030313 (C>A) in the coding region of FLT4 was observed in five GCB cell lines (OCI-LY1, NU-DHL-1, DB, DOHH-2, and WSU-DLCL2), but not in any ABC cell lines. This mutation is associated with the decreased efficacy of the anti-angiogenetic drug sunitinib and reduced progression-free survival in renal cancer patients. A clinical phase II study indicated no effect of sunitinib in relapsed or refractory DLBCL patients and demonstrated higher hematologic toxicity than expected [40]. The gene CYP2B6 encodes for the enzyme belonging to the superfamily of cytochrome P450 monooxygenases. These enzymes are involved in synthesizing cholesterol and steroids and metabolizing anti-cancer drugs such as cyclophosphamide and ifosphamide. From PharmGKB, the efficacy of the chemotherapy drug cyclophosphamide (often used for DLBCL treatment) and the clearance of cyclophosphamide and fludarabine was shown to be affected by the presence of a missense variant at chr19:41515263 (A>G) in the CYP2B6 gene in a study carried out in chronic lymphocytic leukemia (CLL) patients. This mutation was observed in one GCB (WSU-DLCL2) and two ABC (MD903 and SU-DHL-9) cell lines. These results clearly indicate the need for assessing the role of SSVs in appropriately choosing the drugs for targeting various immunogenic pathways.

## 4. Discussion

Immortalized cell lines serve as renewable resources and essential models to study molecular mechanisms underlying cancer development and screening anti-cancer drugs. A comprehensive characterization of their genomes for sequence and structural variations would aid in understanding their potential influence when interpreting biological data. In this study, we carried out identification of small sequence variants (SSVs) using SeqVItA and large structural variations (INDELS, translocations, inversions, MEIs) using TIDDIT, MELT, and CNVs using iCopyDAV across 11 DLBCL cell lines. The validation of predicted SSVs and their prognostic significance in 1295 DLBCL patient samples from cBioPortal for Cancer Genomics is discussed. The analysis revealed the highly heterogeneous genomic landscape of DLBCL. Analysis of the complete spectrum of genetic variants helped in identifying recurrent pathways affected in the ABC and GCB subtypes.

Functional interpretation of a large number of small sequence variations (SNVs and INDELs) identified in whole-genome sequence data is a challenging task. Generally, a majority of these SSVs are not functionally relevant. In this study, we filtered SSVs using the annotation and prioritization module of SeqVItA. In SeqVItA, a variant is assigned ‘high’ priority if the functional impact of the variant score is high (>0.65) from any one of the five resources (SIFT, Polyphen2, MutationTaster, LRT, and PhyloP), clinical association identified in at least one of the three resources (ClinVar, COSMIC, and OMIM), and variant-drug association identified in PharmGKB. A variant is assigned ‘medium’ priority if an annotation is reported from any of the three categories. Mutations identified as high or medium priority were only considered for the analysis. Key cancer genes such as TP53, NOTCH2, and KMT2C were observed to be mutated in the majority of the DLBCL cell lines, irrespective of their subtype. We also identified a few known subtype-specific gene-variants such as STAT6, EZH2, FOXO3 (GCB-specific), and CARD11 and CD79B (ABC specific). Mutations were identified in key genes such as regulator of apoptotic cell death, BCL2 (5/6 cell lines); transcriptional repressor, BCL6 (2/3 cell lines); and transcriptional regulator, TP53 (4/8 cell lines), in agreement with the prior work on these cell lines [17] summarized in Appendix A. Germline mutations observed in the genes FCGR3A and FLT4 have been shown to affect the efficacy of drugs rituximab and sunitinib, respectively, in earlier studies [39,40]. Across all the cell lines, a total of 84 OGs and TSGs were observed to carry mutations (Appendix A), and 41 genes with recurrent mutations were further analyzed. Interestingly, 39/41 genes (except HNF1A and PCSK7) were observed to be mutated in one or more DLBCL patient samples reported in cBioPortal. Several previous studies also indicated DLBCL patients with mutations in some of these 41 genes, namely, MYC, ATM, BCL2, CREBBP, PIM1, and TP53 to be associated with poor survival [41,42]. Of these, patients with mutations in oncogene MYC were associated with worst overall survival (*p*-value < 0.05) on the basis of an analysis of 1295 DLBCL patients reported in cBioPortal. We observed SSVs contributing to impaired development and differentiation of immune cells, maintaining the immunosuppressive environment across all the cell lines via mutations in genes BCL6, CARD11, FANCA, EGFR, FOXO3, PICALM, NF1, TTL, etc. Aberrant lymphoid- and myeloid-derived immune cells influence the immune system by inducing oxidative stress, secreting immunomodulatory factors, regulating amino acid metabolism, and inhibiting T- or NK-cell viability. Subsequently, an immunosuppressive environment promotes development and progression of tumor cells. Another common pathway disrupted through mutations in genes NOTCH2 and NOTCH2NL is the NOTCH signaling pathway across all 11 cell lines. NOTCH signaling is involved in the regulation of cell proliferation, differentiation, and apoptosis, and activation of the NOTCH signaling pathway through genetic variations (mutations and amplifications) is well known in cancer progression. A recent study by Karube et al. [43] showed that mutations in genes participating in the NOTCH signaling pathway triggers the pathway and is associated with poor clinical outcome. Deregulation of cell cycle progression and apoptosis was also observed in all the cell lines. Mutations in various genes were observed to modulate tumor cell proliferation and progression in DLBCL cell lines at various cell cycle transition checkpoints: G0-S (BCL2), G1-S (RUNX1, ATM), G2-M (ATM, PIM2, FANCA), etc. Many of these genes, viz., BCL2, TP53, and PIM2, are also known prognostic markers for DLBCL. The PI3K-AKT pathway is crucial for cell growth and proliferation and was observed to be triggered in a few DLBCL cell lines through mutations in genes GNA13, PIK3R1, and FOXO3. A few subtype-specific alterations in molecular pathways were also observed. For example, chromatic modification through mutations in genes encoding histones and methyltransferases is a common feature observed in the GCB subtype, e.g., mutations in EZH2, CREBBP, and EP300. Similarly, constitutive activation of NF-κB signaling is specific to ABC cell lines. We also observed certain key pathways such as JAK-STAT, RTK, and MAPK in a few cell lines, indicating alternate mechanisms for lymphomagenesis. 

The genomic rearrangement, t(14;18) was reported in six out of seven GCB-subtype DLBCL cell lines (through cytogenetic studies; Appendix A). This translocation involves the enhancer region of IGH partnering with the distal part of proto-oncogene BCL2, as well as subsequent anti-apoptosis of tumor cells. From our analysis, we observed this t(14;18) rearrangement event in four GCB cell lines (OCI-LY1, DB, SU-DHL-6, and DOHH-2) in agreement with the earlier study [44,45,46,47,48]. One possible reason for missing the t(14;18) event in two cell lines (reported in earlier study) was because of the use of a single structural variation detection tool used in this study. It may be noted that some of the novel predictions such as t(6;11), t(3;10), etc., were observed in a number of cell lines (Table 1). Given high sequencing depth and their presence in several cell lines, these are probable true fusion events. However, with computational-based predictions, one major limitation is the inherent biases of the tools, which may result in a few true positives being missed and/or a few incorrect calls (false positives) being made. Obtaining a consensus prediction from multiple variant callers can help in reducing the number of false positives; however, one would miss out many true positives, thereby reducing the recall at the cost of precision. Alternatively, one may consider a union of predictions from various tools that would increase recall but at the cost of precision and should be validated experimentally to filter out true positives. Compared to SSVs and CNVs, functional interpretation of structural variations is a difficult task due to the limited number of studies and dedicated databases. Here, we characterized and derived potential functions of various structural variations through the role of variant-spanning genes from the literature. Translocations in FOXO3/PDGFD and DDX10/SKA3 are interesting due to their possible role in tumor cell growth and survival [49]. Inversions are among the most under-studied genetic variations and pose significant challenges as the consequences of inversions are not clear yet. In our analysis, we observed inversions in genes ABL2 and PRRX1 across all the cell lines. The gene ABL2 (ARG) encodes for a member of the Abelson family of nonreceptor tyrosine kinases and has a role in cell growth and survival. A gene expression study carried out on transformed B-cell lymphomas indicated an increase in ABL2 gene expression upon transformation of follicular lymphoma to DLBCL [50]. We proposed the idea that such a change in expression may result as a consequence of genetic alterations such as inversions, and inversion of ABL2 may act as a biomarker to distinguish DLBCL from other lymphomas. Across the cell lines, we observed a high number of ALU-MEIs compared to other MEIs affecting both OGs and TSGs. These elements were observed to mainly span genes participating in NOTCH signaling, TGF-β signaling, and PI3K-AKT signaling, possibly leading to lymphomagenesis in the cell lines. To validate the presence of these key novel SSVs, CNVs, and SVs, one needs to experimentally confirm in the respective cell lines and corroborate in large data sets with diverse ethnic backgrounds. 

Although cell lines represent stable platforms, it is important to note that cell lines have limitations as a model for studying tumors, and it is essential to consider limitations when interpreting the results. Firstly, cell lines may undergo genetic changes over time as they are propagated in culture. These changes can affect the behavior of the cells and may not accurately reflect the genetic changes present in the original tumor. Secondly, cell lines are usually clonal, i.e., all the cells are genetically identical. This can make it difficult to study the genetic heterogeneity of tumors using cell lines. Finally, cell lines are often derived from a small number of tumors and may not accurately represent the genetic diversity of all tumors of a particular type.

Combining SSVs, CNVs, and SVs helps in identifying a wider spectrum of biological processes and pathways affected. For example, activation of WNT/β-catenin and its crosstalk with the TGF-β signaling pathway as a consequence of SSVs in APC and SMAD4 genes was observed in four DLBCL cell lines that could not be revealed using CNV/SV analysis alone. The combinatorial analysis also revealed possible subgroups within COO subtypes. For instance, several combinations of variants were observed between cell lines of the same subtype, GCB cell lines OCI-LY1 and WSU-DLCL2 (deletion of PTEN, mutations in BCL2, EZH2, FOXO1 inversion, FOXO3/PDGFD translocations), differing from those shared by NU-DHL-1 and DOHH-2 cell lines (deletion of CDKN2A, mutations in CCND3, MYC translocations, NOTCH2 MEIs). Similarly, within cell lines of the ABC subtype, we observed a common set of genetic aberrations in OCI-LY19 and SU-DHL-9 cell lines (mutations in EP300, RUNX1, and TP53). Some combination of genetic variations was also observed in cell lines comprising the two subtypes. For example, cell lines OCI-LY7 (GCB) and MD903 (ABC) shared mutations in APC, BCL6, FANCD2, and SMAD4 and NOTCH2 MEIs, while cell lines SU-DHL-6 (GCB) and NU-DUL-1 (ABC) shared mutations in FOXO3, TP53, and RUNX1 genes; amplification of HSP90AA1 and IRF4 genes; and RAF1 translocations. Thus, we see that the genetic landscape of DLBCL cell lines clearly indicate a distinct evolutionary history of tumor cells within and between COO subtypes, leading to diverse therapeutic responses. Combining all the genetic variations also suggested distinct pathways in DLBCL cell lines, indicating possible mechanisms of lymphomagenesis. For example, cell lines OCI-LY1, WSU-DLCL2, SU-DHL6, and NU-DUL1 were identified with SNVs, CNVs, and SVs participating in the PI3K-AKT signaling pathway, while cell lines NU-DHL-1 and DOHH-2 demonstrated genetic variations altering components of the cell cycle. To understand the translational effect of SSVs, CNVs, SVs, or their combinations in the disruption of immuno-oncogenic pathways, one needs to examine their corresponding mRNA and protein expressions in the cell lines.

Genetic aberrations in key OGs and TSGs also guide in identifying immuno-oncogenic pathways that are targetable. For example, missense mutations in tumor suppressor TP53 were observed in eight DLBCL cell lines (six GCB and two ABC), OCI-LY1, NU-DHL-1, DB, OCI-LY7, WSU-DLCL2, SU-DHL-6, NU-DUL-1, and SU-DHL-9 and in 151/1295 DLBCL patient samples from cBioPortal. They are involved in the regulation of several processes such as cell cycle arrest, induction of apoptosis, DNA repair, and inhibition of angiogenesis and metastasis. Mutations in TP53 were also observed to be associated with drug resistance, poor response to treatment, and short survival of DLBCL patients. The gene PARP1 was identified as accumulating at DNA damage sites and being involved in attracting DNA repair factors. Using PARP1 inhibitors such as olaparib and rituximab, in the presence of TP53 mutations, DNA damage response deficit tumors were shown to be repressed, and increased cytotoxicity was reported in DLBCL cell lines [51]. This suggests that combining PARP1 inhibitors with the standard treatment may lead to improved survival of DLBCL patients with TP53 mutations. Inactivation of APC promotes tumor progression by increasing WNT/β-catenin signaling and was observed in 9/11 DLBCL cell lines (except SU-DHL-6 and NU-DUL-1) and 8/1295 patients in cBioPortal. In normal cells, APC directly interacted with β-catenin and was involved in degrading β-catenin through the formation of a destruction complex. Activation of WNT signaling through loss of function of the APC gene was associated with increased accumulation of β-catenin and its binding to T-cell factor (TCF)/lymphoid-enhancer-binding factor (LEF) in the nucleus. This binding of β-catenin led to remodeling of chromatin structure and activation of genes involved in promoting tumor cell growth and proliferation. Annotations in ClinVar suggest that intronic mutations in the APC gene observed in these eight DLBCL cell lines have been implicated in familial colorectal cancer. The use of tankyrase inhibitors, which downregulate β-catenin activated by mutations in the APC gene, was reported to be an efficient treatment for colorectal cancer patients [52]. In addition to druggable pathways, characterizing SSVs in DLBCL cell lines also identified the key genes FCGR3A, FLT4, and CYP2B6, whose mutations would affect the drug response in the patients. Thus, characterizing the genetic variants not only suggests possible mechanisms of pathogenesis but also provides insights into genome-guided in silico-model-based therapeutic prescriptions. Several targeted therapeutics are already undergoing pre-clinical/clinical testing to efficiently treat DLBCL patients. 

Molecular therapies targeting the aberrant immuno-oncogenic pathways identified in this study can potentially improve the effectiveness of standard R-CHOP. For example, targeting the PI3K-AKT pathway using the PI3K inhibitor duvelisib in combination with R-CHOP was proven to suppress the tumor growth in CHO resistant DLBCL cells [53]. Further, the PI3K inhibitor parsaclisib plus R-CHOP [54], and orally available EZH2 inhibitor tazemetostat plus R-CHOP [55], demonstrated promising use of combinatorial regimens in the early phase clinical trials. Similarly, the orally bioavailable BCL2 inhibitor venetoclax was tested in combination with R-CHOP in NHL patients (including 18 DLBCL patients). The phase I and II trials resulted in complete remission of 79% and 69%, respectively, indicating the addition of venetoclax to R-CHOP in first-line DLBCL treatment [56,57]. In addition, pathways identified in this study such as PI3K-AKT-mTOR signaling inhibited using the drugs idelalisib, AZD8835, and rapamycin; BCR and NF-KB signaling inhibited by ibrutinib and bortezomib can be combined with R-CHOP. Since the targets of these pathways involved in cancer cell survival are distinct from those of R-CHOP, the efficacy of the treatment regimen is expected to be much more efficient. However, more studies are needed to determine the optimal combinations and dosing regimens.

## 5. Conclusions

In our study, we presented an extensive whole-genome sequence analysis of 11 DLBCL cell lines with the anticipation to aid the researchers in inferring meaningful data from these cell lines. Characterizing small sequence variants, copy number, and structural variations suggested causative drivers and enriched pathways leading to lymphomagenesis. Mutations identified in the cell lines were also annotated to infer the efficacy of known and novel drugs employed in DLBCL treatment. Clustering of cell lines on the basis of genetic alterations indicated novel subgroups within and between cell-of-origin subtypes (GCB and ABC). Although an extensive genomic analysis is presented, whether these genetic alterations are translated to transcripts and/or proteins needs to be verified using proteogenomics approaches. Moreover, DLBCL is a highly complex disease and often involves alterations at the epigenomic levels. Thus, integrating the data obtained from multi-omics techniques would enable us in identifying a prioritized set of driver genes and characterizing novel subgroups deviant of the cell-of-origin subtypes.

## Figures and Tables

**Figure 1 cells-12-00596-f001:**
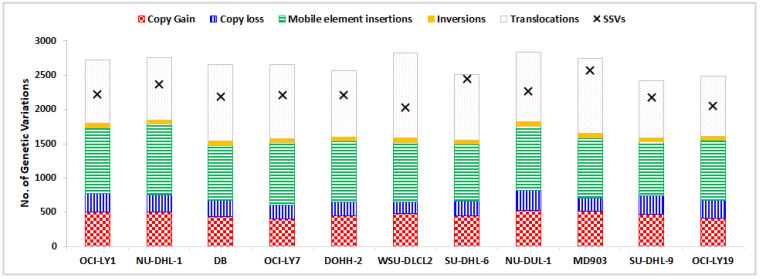
Distribution of five structural variations across DLBCL cell lines, shown as a component bar graph, while the number of SSVs is depicted by ‘×’.

**Figure 2 cells-12-00596-f002:**
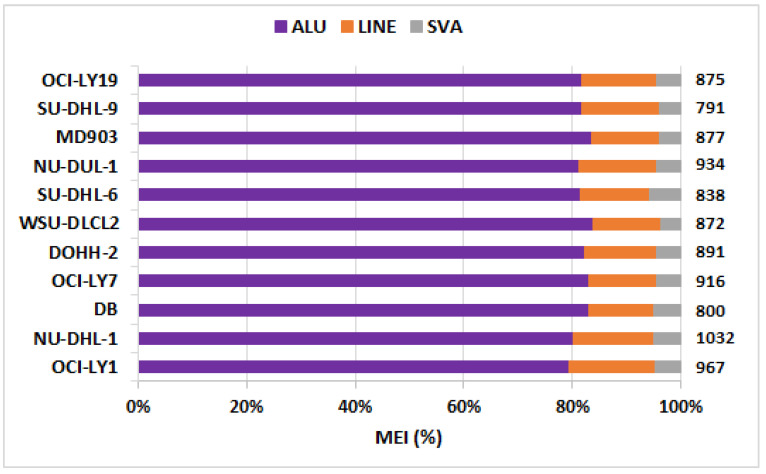
Fraction of three classes of mobile element insertions (MEIs), viz., ALU, LINEs, and SVA. Total number of MEIs per cell line is indicated on the right panel.

**Figure 3 cells-12-00596-f003:**
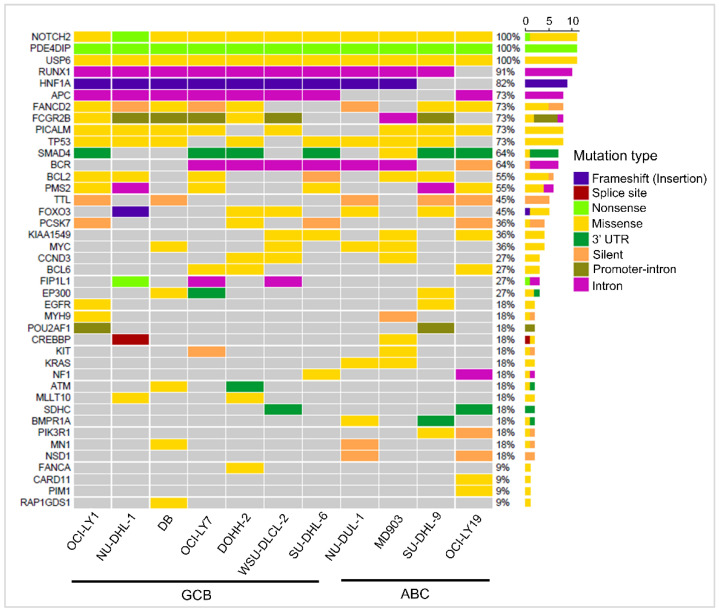
Waterfall plot depicting the mutational landscape of 41 oncogenes and tumor suppressor genes (rows) with SSVs in 11 DLBCL cell lines (columns).

**Figure 4 cells-12-00596-f004:**
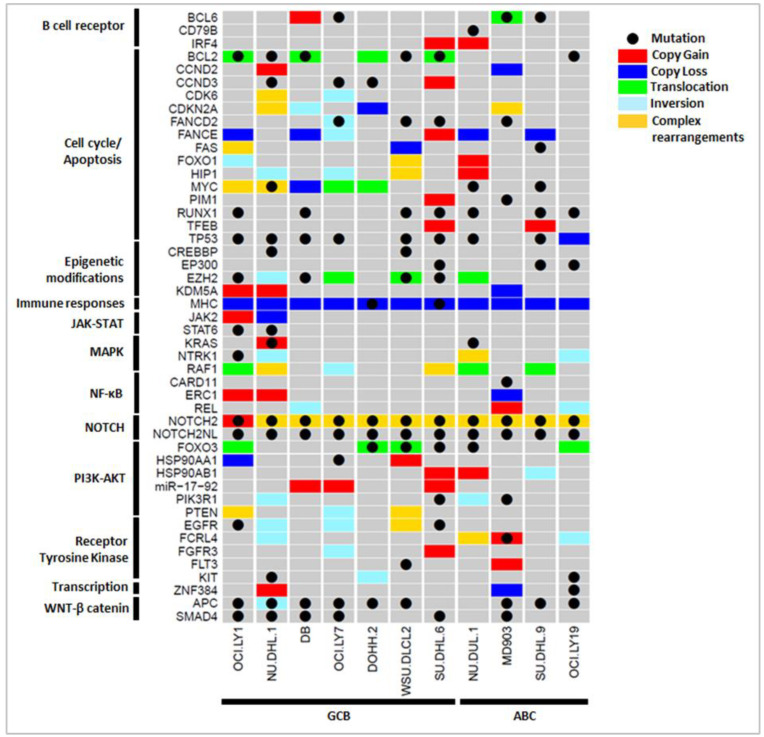
Integration of key SSVs (black dot), CNVs (gain: red, loss: blue), and SVs (translocation: green, inversion: cyan) revealed altered immuno-oncogenic pathways across 11 DLBCL cell lines. Complex genetic rearrangements with ≥2 SVs spanning the gene were also observed (gold yellow).

**Figure 5 cells-12-00596-f005:**
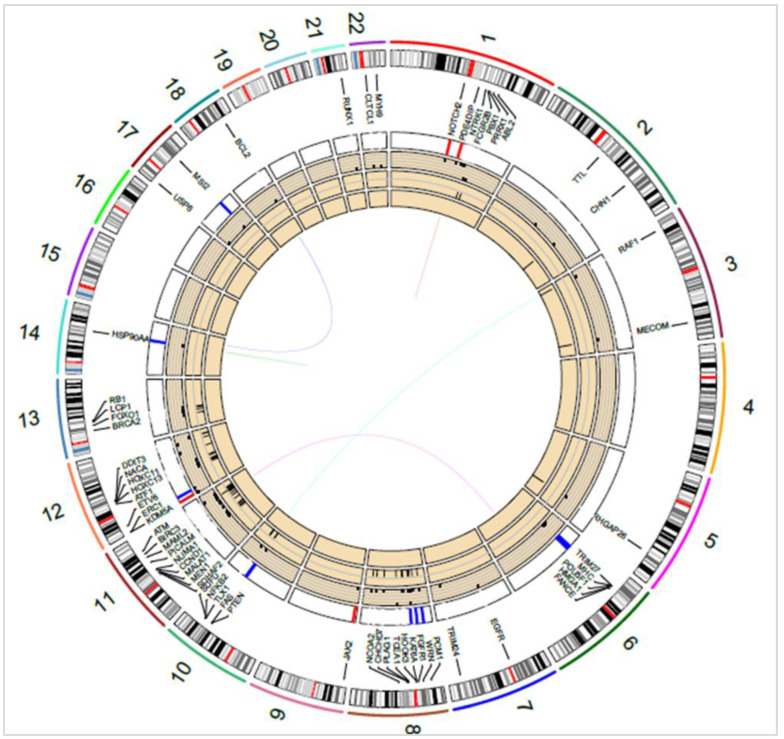
Circos diagram showing key oncogenes and tumor suppressor genes overlapping SSVs, CNVs, and SVs in the OCI-LY1 cell line. Translocations/fusions are shown by connecting lines, inner-to-outer circles: MEIs, INVs, SSVs (dots), and CNVs (red and blue bars). Chromosome numbers 1–22 are indicated next to the outermost circle.

**Figure 6 cells-12-00596-f006:**
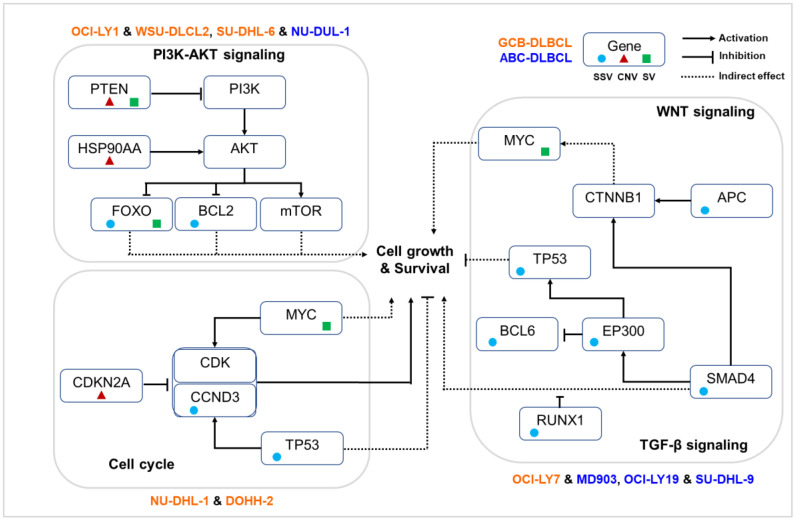
Key pathways disrupted in different sets of DLBCL cell lines according to their shared genetic variations.

**Table 1 cells-12-00596-t001:** Key intra-/inter-chromosomal translocations and associated genes identified in DLBCL cell lines.

Translocation	Gene(s)	Known/Novel	Significance	Cell Line(s)
t(14;18) (q32;q21)	BCL2-IgH	DLBCL (20%)	Proto-oncogenic BCL2 overexpression → inhibits anti-apoptotic pathways	OCI-LY1, DB, DOHH-2, SU-DHL-6
t(3;14) (p27;q32)	BCL6-IgH	DLBCL (30–40%)	B-cell differentiation, survival, cell-cycle control	MD903
t(6;11) (q21;q22)	FOXO3-PDGFD	Novel	Effect cell growth arrest and apoptosis	OCI-LY1, WSU-DLCL2, DOHH-2, OCI-LY19
t(11;13) (q22.3;q12.11)	DDX10-SKA3	Breast cancer	Tumor development through cell apoptosis	NU-DHL-1, DB, SU-DHL-9
t(3;10) (p25.2;q24.2)	RAF1	Novel	MAPK/ERK pathway → cell cycle, migration and differentiation	OCI-LY1, NU-DHL-1, SU-DHL-6, NU-DUL-1, SU-DHL-9
t(7;13) (p15.2;q12.2)	CBX3-FLT1	Novel	Regulate angiogenesis, cell survival, migration	WSU-DLCL2
t(7;11) (p11.2;q13.4)	EGFR-RNF169	Novel	Effect cell proliferation and survival	OCI-LY19
t(5;19) (q33.3;p13.3)	ITK-SHD	Novel	Growth, activation of B cells, T cells	WSU-DLCL2
t(15;2) (q15.1;q11.2)	KNL1-UNC50	Novel	Chromosome segregation	DB
t(7;17) (q21.2;q21.1)	AKAP9-WIPF2	Novel	G2/M transition of mitotic cell cycle	NU-DHL-1
t(7;16) (q36.1;q21)	EZH2	Novel	GC B cell development, catalyzed H3K27 methylation	OCI-LY7, WSU-DLCL2, NU-DUL-1
8q24.21	MYC	DLBCL	G0/S1 phase, downregulating cell cycle inhibitors	NU-DUL-1, OCI-LY7, DOHH-2
1q21.1	PDE4DIP	Novel	Cell migration, mitotic spindle orientation, cell-cycle progression	OCI-LY1, WSU-DLCL2, DB, NU-DUL-1
14q24.1	RAD51B	Uterine leioyoma	DNA repair by homologous recombination	OCI-LY1
13q13.3	LHFPL6	Novel	Member of lipoma HMGIC fusion partner gene family	NU-DUL-1
16q24.3	CBFA2T3	Novel	Transcriptional repressor	NU-DHL-1, SU-DHL-6, OCI-LY19
7q31.1	FOXP2	Novel	Chromosomal instability → cancer initiation and progression	WSU-DLCL2

**Table 2 cells-12-00596-t002:** List of key genes altered as a result of MEIs in 11 DLBCL cell lines.

Region	Gene	Significance	DLBCL Cell Lines
1p12	NOTCH2	NOTCH signaling pathway	ALU: NU-DHL-1, DB, NU-DUL-1, OCI-LY7, MD903, SU-DHL-9, DOHH-2, WSU-DLCL2, SU-DHL-6
2q31.1	CHN1	GPCR signaling	ALU: OCI-LY1, NU-DHL-1, DB, SU-DHL-9, WSU-DLCL2, SU-DHL-6
2p23.2	ALK	Cancer cell growth	ALU: NU-DHL-1, DOHH-2, WSU-DLCL2, OCI-LY19
3p25.2	RAF1	MAPK/ERK pathway → cell cycle, migration and differentiation	SVA: OCI-LY1, NU-DHL-1, SU-DHL-9, SU-DHL-6
3q26.3	MECOM	TGF-β signaling → regulating proliferation, apoptosis	ALU: OCI-LY1, DB, DOHH-2, WSU-DLCL2
11p13	WT1	Tumor suppressor, cellular development and survival	ALU: NU-DHL-1, DB, MD903, OCI-LY19
14q24.1	RAD51B	Overexpression leads to delay in cell cycle G1	ALU: SU-DHL-9, SU-DHL-6, OCI-LY19
10p12.1	ABI1	Regulation of EGF-induced Erk pathway activation	ALU: MD903, SU-DHL-9
2q11.2	AFF3	Lymphoid development and oncogenesis	ALU: NU-DUL-1, OCI-LY7
3q28	LPP	Cell adhesion, cell shape, and motility maintenance	ALU: OCI-LY19, LINE: MD903
7q31.2	MET	RAS-ERK, PI3K-AKT pathway	ALU: DOHH-2
9p21.3	MLLT3	Histone modifications	ALU: MD903
9q22.33	NR4A3	Proliferation of vascular smooth muscle, myeloid progenitor cell	ALU: OCI-LY7
1q21.1	PDE4DIP	Cell migration, mitotic spindle orientation, cell-cycle progression	ALU: OCI-LY7
20q13.33	SS18L1	Subunit of a neuron-specific chromatin-remodeling complex	ALU: WSU-DLCL2

**Table 3 cells-12-00596-t003:** Probable subgroups within and between DLBCL subtypes observed in cell lines based on shared genetic variations.

Cell Line Pairs	Shared Genetic Variations	Key Pathway
OCI-LY1 (GCB) and WSU-DLCL2 (GCB)	Mutations in EZH2 and BCL2, loss and inversion in PTEN, FOXO1 inversion, FOXO3-PDGFD translocations	PI3K-AKT signaling
NU-DHL-1 (GCB) and DOHH-2 (GCB)	Mutations in CCND3, loss of CDKN2A, MYC translocations, NOTCH2 MEIs	Cell cycle
OCI-LY19 (ABC) and SU-DHL-9 (ABC)	Mutations in EP300, RUNX1, TP53	TGF-β signaling
OCI-LY7 (GCB) and MD903 (ABC)	Mutations in APC, BCL6, FANCD2, SMAD4, NOTCH2 MEIs	TGF-β-WNT signaling
SU-DHL-6 (GCB) and NU-DUL-1 (ABC)	Mutations in FOXO3, TP53, RUNX1, gain of HSP90AA1 and IRF4, translocations in RAF1	PI3K-AKT signaling

## Data Availability

Not applicable.

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
