# Peer review of "In Silico Identification and Functional Characterization of Genetic Variations across DLBCL Cell Lines"

_cells, 2023, doi:10.3390/cells12040596_

Round 1
Reviewer 1 Report
The authors profiled major genetic alterations in 11 DLBCL cell lines and found some common or cell line specific oncogenic mutations. Some key oncogenic signaling pathways resulted from these genetic alterations were discussed, including PI3K-AKT, NOTCH, WNT, etc. In general, the manuscript was well-written, the data are straightforward and convincing, which may yield new insights into the new targeted therapy for DLBCL in the future. The strength of this manuscript might be enhanced if the following points will be addressed by the authors:
1. Fig. 3. the quality of this figure must be improved by providing high resolution gene names on the left side of the "% mutant" panel.
2. Please provide some biochemical/biological validations for the enhanced AKT, NOTCH or TGFb signaling in the selected cell lines.
3. Limitations on the data collected from cell lines, rather than the clinical patients samples, were not discussed. Due to culturing and passaging conditions in in vitro, the possibilities of non-relevant genetic alterations in the cells lines couldn't be excluded.
4. It'll be highly appreciated to expand discussions on the therapeutic implications of these findings from this study to improve the efficacy of the current CHOP/R-CHOP regimen.
Author Response
A point-by-point response is provided in the uploaded file.

Reviewer 2 Report
The authors present their whole-genome sequencing study on 11 Diffuse Large B-cell Lymphoma (DLBCL) cell lines. Specifically, the authors characterized different types of mutations, such as SSVs, CNVs, INVs, TRAs, and MEIs and their subcategories, in the 11 DLBCL cell lines. They found that genetic variations in the cell lines tend to disrupt key immuno-oncogenic pathways, despite different routes of tumorigenesis occurring in individuals. In addition, the authors depicted comprehensive roles of genetic variations in lymphomagenesis and diverse paths of DLBCL pathogenesis in the paper, providing valuable insights for research and treatments. Despite some minor issues, the study would significantly contribute to DLBCL studies.
1. Several sentences in the first paragraph of the Introduction section lack references.
a. Line 31: “Several classification systems” needs references.
b. Line 34: “The Ann Arbor staging classification system.”
c. Line 36: “IPI”
d. Line 38-39: “Gene expression profiling of DLBCL patients has led to the identification of two distinct subtypes based on the cell-of-origin (COO) classification, Germinal centre B-cell (GCB) and Activated B-cell (ABC) subtypes.”
2. Line 215-217: “Interestingly, from the data in cBioPortal, mutations in CARD11 and PIM1 tend to co-occur in 29/1295 DLBCL patients with high significance (p-value: 0.03)”. It is unclear how the p-value was calculated and which statistical method was used.
Author Response
A point-by-point response to the reviewer is provided in the uploaded file.
